# Maternal circulating miRNAs contribute to negative pregnancy outcomes by altering placental transcriptome and fetal vascular dynamics

**Marisa R. Pinson**[1], **Alexander M. Tseng**[1], **Tenley E. Lehman**[1], **Karen Chung**[1], **Jessica Gutierrez**[2], **Kirill V. Larin**[2], **Christina D. Chambers**[3,4], **Rajesh C. Miranda**[1,5]*, **CIFASD**[¶]

1 Department of Neuroscience and Experimental Therapeutics, Texas A&M University Health Science Center, Bryan, TX, United States of America, 2 Department of Biomedical Engineering, University of Houston, Houston, TX, United States of America, 3 Clinical and Translational Research Institute, University of California San Diego, San Diego, CA, United States of America, 4 Department of Pediatrics, University of California San Diego, San Diego, CA, United States of America, 5 Women's Health in Neuroscience Program, Texas A&M University Health Science Center, Bryan, TX, United States of America

¶ Membership of the CIFASD; Collaborative Initiative on Fetal Alcohol Spectrum Disorders is provided in the Acknowledgments.
* rmiranda@tamu.edu

**Data Availability Statement:** Data relevant to this study are included in the Supporting Information

## Abstract

Circulating miRNAs the in blood are promising biomarkers for predicting pregnancy complications and adverse birth outcomes. Previous work identified 11 gestationally elevated maternal circulating miRNAs (HEamiRNAs) that predicted infant growth deficits following prenatal alcohol exposure and regulated epithelial–mesenchymal transition in the placenta. Here we show that a single intravascular administration of pooled murine-conserved HEa-miRNAs to pregnant mice on gestational day 10 (GD10) attenuates umbilical cord blood flow during gestation, explaining the observed intrauterine growth restriction (IUGR), specifically decreased fetal weight, and morphometric indices of cranial growth. Moreover, RNA-seq of the fetal portion of the placenta demonstrated that this single exposure has lasting transcriptomic changes, including upregulation of members of the Notch pathway (*Dll4*, *Rfng*, *Hey1*), which is a pathway important for trophoblast migration and differentiation. Weighted gene co-expression network analysis also identified chemokine signaling, which is responsible for regulating immune cell-mediated angiogenesis in the placenta, as an important predictor of fetal growth and head size. Our data suggest that HEamiRNAs perturb the expression of placental genes relevant for angiogenesis, resulting in impaired umbilical cord blood flow and subsequently, IUGR.

## Introduction

MicroRNAs (miRNAs) are non-coding RNAs, approximately 20–24 nucleotides in length, that regulate gene expression networks by targeting messenger RNAs for degradation or by

files, and found at NCBI GEO database at accession number GSE190017.

**Funding:** ). This work was supported by grants from the NIH, R01 AA024659 (RCM), R01 HD086765 (KVL,RCM), U01 AA014835 (CC), F30 AA027698 (MRP), and F31 AA026505 (AMT). The funders had no role in study design, data collection and analysis, decision to publish, or preparation of the manuscript.

**Competing interests:** The authors have declared that no competing interests exist.

blocking translation [1, 2]. miRNAs are found in circulation, chaperoned by EVs [3, 4], lipoproteins [5], Argonaute 2 [6], or other RNA-binding proteins [7]. Moreover, these circulating miRNAs are capable of altering gene expression in recipient cells [4, 5, 8], and consequently may be classified as endocrine molecules. The routine availability of blood samples for monitoring patient health means that these endocrine miRNAs are candidate predictors of disease risk and severity, and specifically, for monitoring pregnancy health and birth outcomes. For instance, several studies have identified potential miRNA biomarkers for pregnancy outcomes such as pre-eclampsia [9, 10], preterm birth [11] and fetal growth [12, 13]. A few research groups, including ours, have shown that in both animal models [14] and human studies [15–18], circulating miRNAs are biomarkers for prenatal alcohol exposure, as well as a predictors of birth outcomes and infant neurodevelopment. These studies advanced the possibility that circulating miRNAs could serve as predictive biomarkers to assist with the diagnosis of Fetal Alcohol Spectrum Disorders (FASD), which are leading cause of developmental disabilities, prevalent in ~1.1 to 9.8% of school-aged children in the US [19].

Specifically, our previous study in a prospective cohort of pregnant women in western Ukraine assessed maternal miRNA profiles in second and third trimesters in women who were heavily exposed to alcohol during pregnancy and subsequently had adverse infant growth and neurodevelopmental outcomes (HEa mothers). We compared these miRNA profiles to mothers who were equally heavily exposed to alcohol but whose infants were unaffected at birth (HEua mothers) and control, un-exposed (UE) mothers. These studies identified a group of 11 maternal miRNAs, hsa-miR-222-5p, hsa-miR-187-5p, hsa-miR-299-3p, hsa-miR-491-3p, hsa-miR-885-3p, hsa-miR-518f-3p, hsa-miR-760, hsa-miR-671-5p, hsa-miR-449a, hsa-miR-204-5p, and hsa-miR-519a-3p, that were stably elevated in both second and third trimesters in plasma of HEa mothers, but not HEua or UE mothers [18]. Moreover, these 11 miRNAs ($_{HEa}$miRNAs) collectively explained between 24 and 31% of the variance in infant growth parameters at birth [20]. The fact that these miRNAs were elevated in HEa but not HEua mothers implies that these miRNAs were not specifically sensitive to prenatal alcohol exposure, but rather, were a collective and early 'danger signal' for an adverse infant birth outcome. Indeed, a review of the literature indicated that a majority of these $_{HEa}$miRNAs are also elevated in other pregnancy-related complications, including fetal growth restriction and pre-eclampsia [20], suggesting that these growth deficit-predicting maternal miRNAs may also mediate common underlying mechanisms.

Further studies also showed that these 11 $_{HEa}$miRNAs collectively, but not individually, inhibited invasiveness and epithelial-mesenchymal transition (EMT) in human trophoblast cell lines. The EMT pathway is critical for proper placental development and placental vascular function [21–24], and therefore, inhibiting EMT is predicted to deprive the fetus of nutrition and contribute to intrauterine growth restriction (IUGR). Consistent with this prediction, we found that, in a murine model of pregnancy, a single prenatal exposure to a core set of 8 $_{HEa}$miRNAs that were conserved in placental mammals ($m_{HEa}$miRNAs), resulted in IUGR and placental deficits [20], pointing to the fact that at least this core group of $m_{HEa}$miRNAs may mediate placental and fetal growth inhibition, rather than simply serving as biomarkers. For this reason, we interrogated the effects of prenatal combined exposure to these 8 mammalian-conserved $m_{HEa}$miRNAs on fetal vascular dynamics and the placental transcriptome and examined the relationship between these measurements and fetal anatomical and morphometric outcomes.

## Materials and methods

All procedures were performed in accordance with Texas A&M Institutional Animal Care and Use Committee guidelines and approval. C57/BL6NHsd dams and male mice (Envigo) were

bred in house. All procedures were approved by the Texas A&M Institutional Animal Care and Use Committee (IACUC). We also followed the ARRIVE guidelines for reporting results. General study methodologies and paradigms are as previously published [25].

## Mouse model for HEamiRNA overexpression

As previously published, nulliparous C57/BL6NHsd dams were injected on gestational day 10 (GD10) via tail vein, with either 50 μg of miRNA miRVana mimic negative control (Cat No. 4464061; Thermo Fisher Scientific) or pooled murine-conserved $m_{HEa}$miRNA miRVana mimics in In-vivo RNA-LANCEr II (3410–01; Bioo Scientific), according to the manufacturer's instructions [20]. The 50 μg of pooled $_{HEa}$miRNA mimics consisted of equimolar quantities of mmu-miR-222-5p, mmu-miR-187-5p, mmu-miR-299a, mmu-miR-491-3p, miR-760-3p, mmu-miR-671-3p, mmu-miR-449a-5p, and mmu-miR-204-5p mimics. At GD18, pregnancies were terminated and tissue was snap frozen in liquid nitrogen and stored at −80˚C preceding RNA isolation.

## Blood flow imaging

As previously published [25, 26], C57/BL6NHsd dams were anesthetized using isoflurane (3–4% for initiation of anesthesia, 1% for maintenance), and maintained supine on a temperature-controlled mouse platform (with sensors for monitoring of maternal electrocardiogram, respiration, and core body temperature; Fuji/Visualsonics, Toronto, Canada). Maternal temperature was maintained at 34–37˚C and maternal heart rate at ~425 beats/minute by adjusting the level of anesthesia. The abdomen was shaved and depilated (using Nair) to improve contact with the transducer. Pre-warmed (37˚C) ultrasound gel (Aquasonic, Parker Laboratories Inc., NJ) was applied to the dam's abdomen prior to positioning the transducer. An initial scan was performed on both uterine horns to verify the number and location of all fetuses. For each pregnant dam, two fetuses (one at the end of the left uterine horn and one at the end of the right uterine horn) were selected for both pre- (GD10) and post-tail vein injection (GD12, GD14, GD18) treatment scans. The same fetuses from each uterine horn were monitored throughout gestation. Both color and pulse wave Doppler measurements for umbilical arteries and ascending aorta were obtained using a high-frequency VEVO2100 ultrasound imaging machine coupled to an MS550D Microscan™ transducer with a center frequency of 40MHz (Visualsonics, Canada).

Data from pulse wave Doppler imaging experiments were analyzed using the VEVO2100 measurement and analysis software (Fuji/Visualsonics, CA) to assess Acceleration (in mm/sec$^2$) and Velocity-Time Integral (VTI, in mm$^3$/sec). Acceleration is defined as the change in blood flow velocity over time from the onset of systolic forward flow to peak velocity [27]. Acceleration can be used as a measure of cardiac output in peripheral vessels [28] in human patients and in embryonic mice studies [29] and correlates with arterial resistance. VTI, the area under the velocity envelope [27] is a substitute measure of cardiac stroke volume through a specific blood vessel. Acceleration and VTI measurements were obtained from the umbilical artery and fetal ascending aorta. Each data point represents measurements from one fetus from either the left or right uterine horn, in one pregnant dam (to eliminate litter and uterine position effects). Results presented are mean ± SEM for each group (n = 5–6 pregnant dams), normalized to the average baseline value for the control, pre-$m_{HEa}$miRNA exposed group.

## Optical coherence tomography and image analysis

Imaging was performed using a swept source optical coherence tomography (SSOCT) system, which consists of a broadband laser (Santec Corporation, Japan) with a central wavelength of

1310 nm, sweep rate of 50kHz, and axial resolution of 9.76 μm. All images were processed using Matlab, and Amira 5.4 was used to analyze and measure dimensions of GD18 fetal mouse limbs and philtrum length.

## RNA isolation and RNA library preparation and sequencing

Total RNA from the non-decidual (labyrinthine with the junctional zone) portion of mouse placentas was isolated using the miRNeasy mini kit (Qiagen; Catalog # 217004). Prior to analysis, RNA quality was assessed using an Agilent TapeStation RNA assay. Total RNA concentration was quantified via Qubit Fluorometric assay, and subsequently, all samples were normalized to an equivalent starting concentration. Sequencing libraries were prepared using the TruSeq Stranded Total RNA with Ribo-Zero Library Preparation kit (Illumina; San Diego, CA). Each sample was uniquely indexed (barcoded) to allow for the pooling of all samples in a single sequencing run. Library size and quality were then assessed with an Agilent TapeStation D1000 DNA assay. Samples were normalized to ~4nM and pooled equally. Sequencing was performed on an Illumina NovaSeq S4 Flow Cell, XP running with a 150 cycle, paired-end (2x150) sequencing run to generate approximately 50 million read pairs per sample.

## Bioinformatic analysis

Raw RNA-sequence data were analyzed to identify significant differences in gene expression between the tail vein control vs. $m_{HEa}$miRNA treatment groups. All reads were evaluated and trimmed of all adapter sequences and low-quality bases using Trimmomatic read trimmer [30]. Using Trimmomatic and the corresponding adapter sequences file for Illumina, reads were scanned with a sliding window of 5, cutting when the average quality per base drops below 20, then trimming reads at the beginning and end if base quality drops below 25, and finally dropping reads if the read length is less than 35. Reads were then mapped to the Mus musculus (mm10) genome assembly. Read mapping was performed using HISAT2 genomic analysis software platform version 2.1.0 [31]. Transcript-wise counts were generated using HTSeq [32]. Differential gene expression tests were then performed using DESeq2 software version 2.1.8.3 following the guidelines recommended by Love and colleagues [33] using a 'treatment' x 'sex' experimental design. A total of 22,103 genes had at least one read count in at least one sample and were processed for differential expression analysis using the regularized logs of normalized gene counts derived from DESeq2. We did observe a few genes whose expression was sex-dependent (see S1 and S2 Tables). However, there were only 6 genes identified as uniquely significant in male placentas and 1 gene as uniquely significant in female placentas. Therefore, our analyses focused on main effect of treatment on differential gene expression. All analyses were done on the Galaxy instance of the TAMU HPRC (https://hprcgalaxy.tamu.edu/).

Correlations were calculated using the cor() function in R, and partial correlations were calculated using the *ppcor* package in R [34]. Correlation plots were created with the *corrplot* R package [35]. The 'R' based *EnhancedVolcano* package was used to make the volcano plot [36]. Pathway analysis was conducted using *ReactomePA* [37] on the differentially regulated genes (FDR adjusted p-value < 0.05). *ReactomePA* utilized the KEGG database [38] and the *Pathview* R package [39] to visualize differentially regulated pathways. Weighted gene co-expression network analysis (WGCNA) was conducted using the *WGCNA* R package [40] to construct networks of relatedness and identify eigengenes, or 'hub' genes, from gene expression data.

To gain a better understanding of which cell types of the non-decidual zone are most sensitive to $_{HEa}$miRNA exposure on GD10, we leveraged publicly available single cell RNA-seq (scRNA-seq) resources. We and others have reported similar analyses using cell-type-specific

markers obtained from scRNA-seq studies in order to extrapolate cell composition of tissue used in bulk RNA-seq [41, 42]. Cell cycle associated (S and G2/M phase) gene lists were extracted from the 'R' *Seurat* package [43] and applied to our bulk RNA-seq dataset. We calculated z-scores across individual genes on the extracted gene lists and then averaged these individual gene z-scores within each sample. The average z-score of each sample was then used in 2-way ANOVA analyses (sex x treatment) completed in "R". This strategy allowed us to compare relative expression of differentially expressed genes (e.g., housekeeping genes are more highly expressed than other cell type specific genes) because our interest is not in absolute expression levels but rather relative expression levels as a whole for each gene list (e.g., do all genes go up or down due to treatment). This process was repeated for analysis of caspase transcripts (apoptosis-related caspases, including Caspase 2 (CASP2), CASP3/6/7/8/9/10 and the inflammation-related caspases, CASP1/4/5/11/12) [44] and transcripts for cell-type-specific marker genes [45].

## Statistical analysis

Statistical analyses were conducted using the GraphPad Prism software, version 9.2.0 for Windows. Results are expressed as the mean ± SEM. The overall group effect was analyzed for significance using two-way ANOVA with Tukey's HSD post hoc testing when appropriate (i.e., following a significant group effect or given a significant interaction effect between experimental conditions in two-way ANOVA), to correct for a family-wise error rate. All statistical tests, sample sizes, and post hoc analysis are appropriately reported in the results section. A value of $P < 0.05$ was considered statistically significant and a value of $0.1 < P < 0.05$ was considered as trending towards significance

## Results

### Prenatal $m_{HEa}$miRNA exposure attenuates placentally directed blood flow

A schematic diagram of the general study protocol is as outlined in Fig 1A. Eight out of 11 $_{HEa}$-miRNAs were evolutionarily conserved in placental mammals and 3 were primate and/or human specific. Moreover, our previous study identified these 8 mammalian-conserved $_{HEa}$-miRNAs ($m_{HEa}$miRNAs) as collectively sufficient in inducing IUGR in a mouse model [20]. These $m_{HEa}$miRNAs are hypothesized to constitute the conserved core of maternal 'danger signal' that predicts growth restriction. We therefore examined the effect of a single exposure to these eight $m_{HEa}$miRNAs (50 μg of pooled equimolar quantities of mmu-miR-222-5p, mmu-miR-187-5p, mmu-mir-299a, mmu-miR-491-3p, miR-760-3p, mmu-miR-671-3p, mmu-miR-449a-5p and mmu-miR-204-5p mimics [20]) on gestational day 10 (GD10) fetal blood flow in the umbilical artery and ascending aorta, as measured by the parameter Velocity-Time Integral (VTI, in mm$^3$/sec), which measures fetal cardiac systolic output within those blood vessels (Fig 1B). Our data show that while cardiac output did not change in ascending aorta over development (GD12 to GD18, $F_{(2, 40)} = 1.767$, $p = 0.184$; Fig 1C), it significantly increased, by ~1.8-fold in the umbilical artery over the same time period ($F_{(2, 40)} = 17.85$, $p = 0.000003$; Fig 1D), suggesting that the umbilical artery preferentially experiences increased blood flow volume during cardiac systole, coincident with fetal development. $m_{HEa}$miRNAs also did not affect VTI in the ascending aorta ($F_{(1,20)} = 0.424$, $p = 0.522$; Fig 1B), but did result in a trend towards a diminished elevation between GD12 to GD18 (main effect of treatment, $F_{(1,20)} = 3.208$, $p = 0.088$; Fig 1C). A planned post-hoc comparison between control and $m_{HEa}$miRNAs-exposed groups at GD18, showed that cardiac output through the umbilical cord in the $m_{HEa}$-miRNA -exposed group was ~14.3% less than that in the control umbilical cord ($t_{(20)} = 2.334$, $p = 0.03$; Fic 1c, see S1 Data) just prior to parturition. In contrast, during this time period,

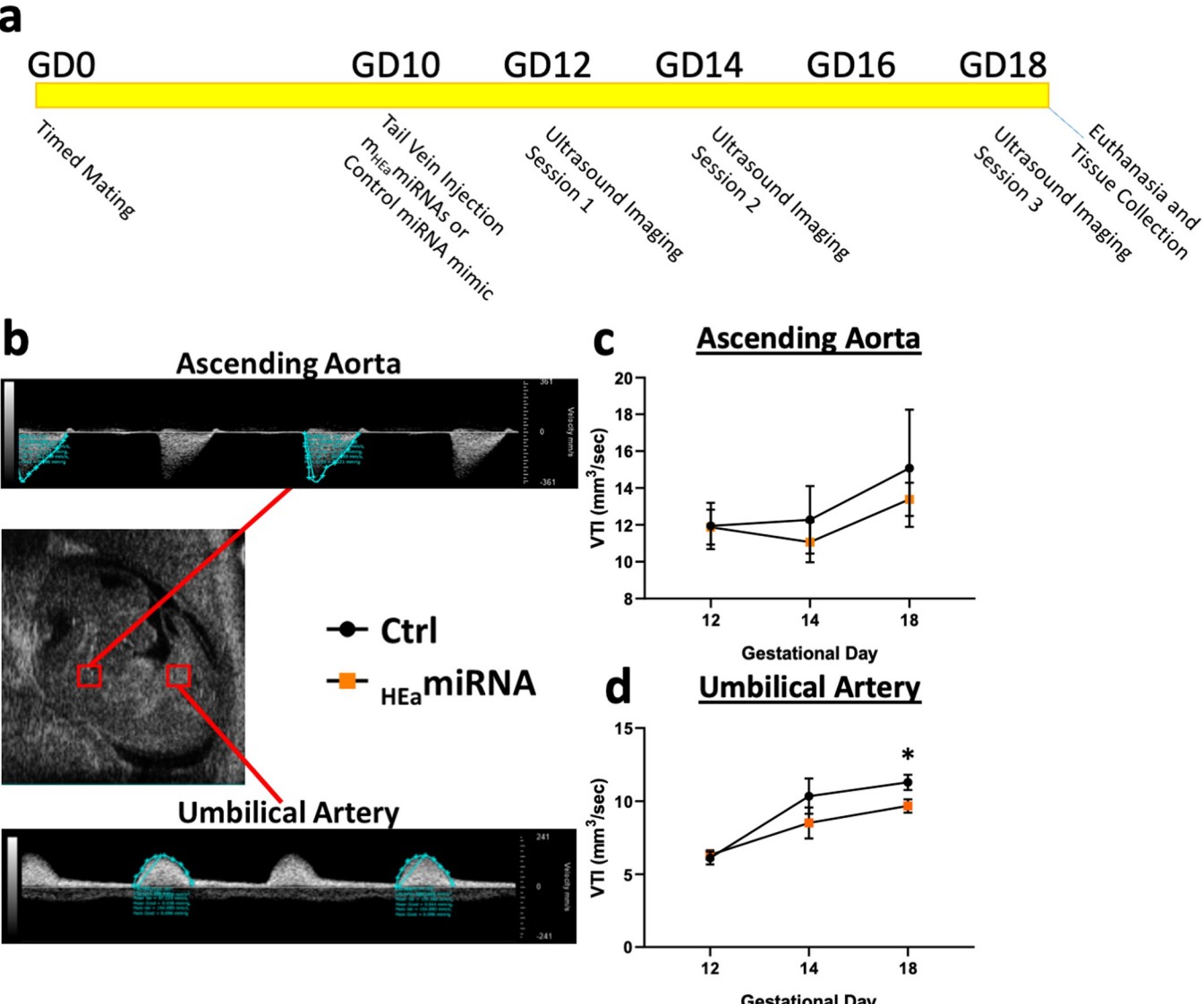

**Fig 1. Prenatal $m_{HEa}$miRNA exposure attenuate placentally directed blood flow.** (**a**) Schematic of study timeline. (**b**) Representative mid-sagittal image of fetal mouse and measured blood flow of ascending aorta and umbilical artery. Time (in seconds) is represented on 'X' axis in each trace, and the 'Y' axis indicates Velocity (mm/sec). The blue envelope in each trace represents the area under the curve and is used to calculate the Velocity Time Integral (VTI). (b,c) Graphs depict VTI (mm³/sec, 'Y' axis) over gestational age ('X') axis of the ascending aorta (**c**) and umbilical artery (**d**) in fetal mice. Timed-pregnant dams were exposed to sham control and $m_{HEa}$miRNA treatments on GD10, and fetal blood flow observed on GD12, 14 and 18. Results are expressed as the mean ± SEM, Control n = 10, $m_{HEa}$miRNA n = 12. Main effect of Treatment *p < 0.05.

acceleration through the umbilical artery, a measure of arterial resistance, was not changed either by gestational age or by $m_{HEa}$miRNA exposure (all p-values >0.05, S1 Fig).

## Prenatal $_{HEa}$miRNA exposure does not impair fetal limb growth and philtrum development

Previously, we found that prenatal $m_{HEa}$miRNA exposure resulted in impaired fetal growth and placental size [20]. Specifically, fetal weight, placental weight, crown-rump length (CRL), snout-occipital distance (SOD), and biparietal diameter (BPD) were all decreased in fetuses prenatally exposed to $m_{HEa}$miRNAs regardless of sex. Therefore, we were interested in

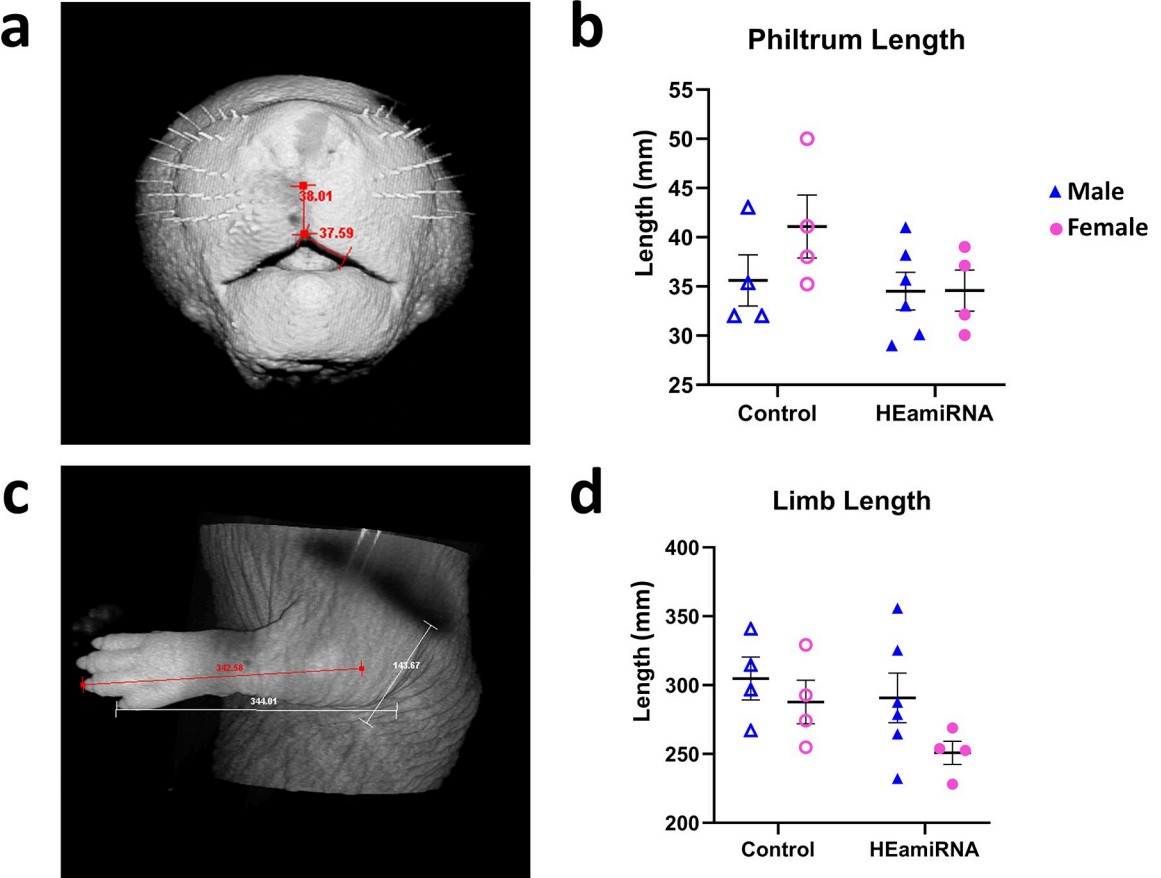

**Fig 2. Prenatal $m_{HEa}$miRNA exposure does not alter facial morphogenesis and long bone formation.** Representative images of GD18 fetal mouse philtrum length (**a**) and limb length (**c**). Quantification of philtrum length (**b**) and limb length (**d**). Control male n = 4, Control female n = 4, $m_{HEa}$miRNA male n = 6, $m_{HEa}$miRNA female n = 4.

determining the extent to which other diagnostic features of FASD, facial morphogenesis, as measured by philtrum length (Fig 2A) and long bone development, as measured by limb length (Fig 2C), might also be sensitive to *in utero* $m_{HEa}$miRNAs exposure. However, we observed no effect of treatment or sex on philtrum length (main effect of treatment, $F_{(1,14)}$ = 1.285, p = 0.276; main effect of sex, $F_{(1,14)}$ = 2.407, p = 0.143; Fig 2B) or limb length (main effect of treatment, $F_{(1,14)}$ = 2.410, p = 0.143; main effect of sex, $F_{(1,14)}$ = 3.006, p = 0.105; Fig 2D).

## Prenatal $m_{HEa}$miRNA exposure alters the relationship between blood flow and growth outcomes

We examined the relationships between fetal-placental growth parameters and blood flow dynamics by first completing Pearson correlation analysis that did not account for prenatal $m_{HEa}$miRNA exposure. Significant positive correlations were identified between CRL and GD18 VTI of the umbilical artery (Pearson r = 0.623, p = 0.041) and between SOD and the age-related change in VTI (ΔVTI GD12->14) of the umbilical artery (Pearson r = 0.684, p = 0.020) (Fig 3A; S2A Fig, see S2 Data). Additionally, several significant positive relationships were identified between philtrum length and GD18 VTI of the umbilical artery (Pearson r = 0.783, p = 0.022), ΔVTI GD12->18 of the umbilical artery (Pearson r = 0.793, p = 0.019), and ΔVTI GD12->14 of the umbilical artery (Pearson r = 0.791, p = 0.020). After correcting

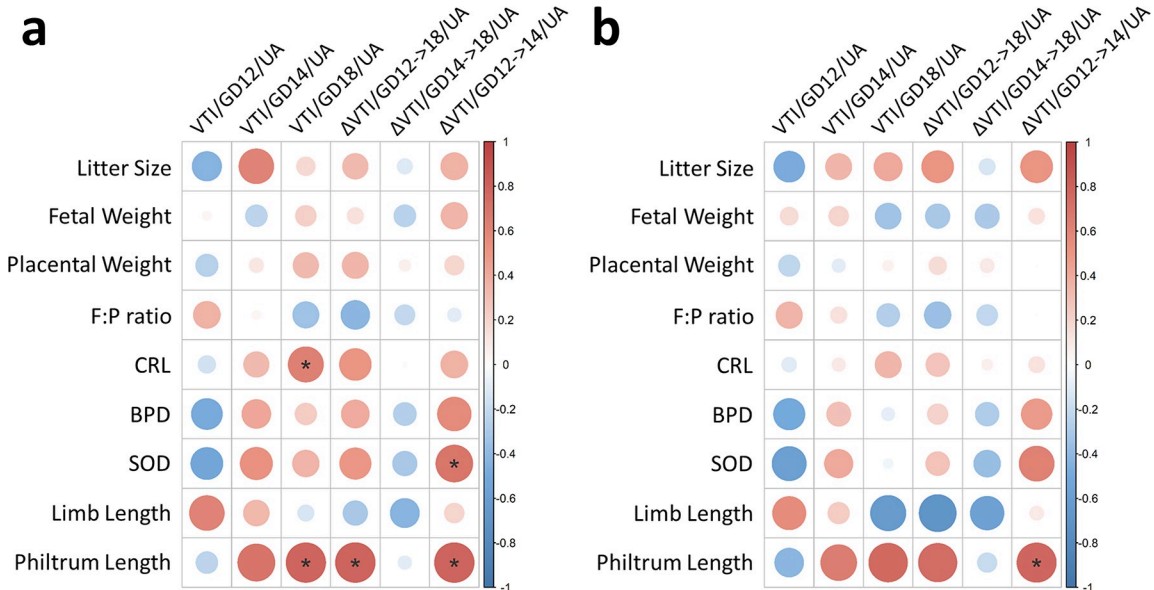

**Fig 3. Prenatal $m_{HEa}$miRNA exposure alters relationship between blood flow and fetal growth outcomes.** Heatmap of Pearson correlation matrices for uncorrected analyses (**a**) and analyses correcting for prenatal $m_{HEa}$miRNA exposure (**b**). For all correlation matrices, color intensity and dot size indicate a stronger correlation, with red representing a positive relationship and blue representing a negative relationship. BPD = biparietal diameter. CRL = crown-rump length. F:P ratio = placental efficiency. SOD = snout-occipital distance. VTI = Velocity Time Integral. UA = umbilical artery. Control male n = 4, Control female n = 5, $m_{HEa}$miRNA male n = 6, $m_{HEa}$miRNA female n = 4. *$p < 0.05$.

for prenatal $m_{HEa}$miRNA exposure using partial correlation analysis, all of these significant relationships disappeared except for the relationship between philtrum length and ΔVTI GD12->14 of the umbilical artery (Pearson r = 0.776, p = 0.040) (Fig 3B; S2B Fig), demonstrating that prenatal $m_{HEa}$miRNA exposure mediates at least some of the relationships between placentally directed blood flow and fetal growth metrics.

## Prenatal $m_{HEa}$miRNA exposure induces differential gene expression in the placenta and alters Notch signaling pathway

Up- and down-regulated genes were identified in non-decidual (labyrinthine with junctional zone) portions of placentas that were common to both male and female fetuses (i.e., sex-independent, males n = 4/control and n = 6/$m_{HEa}$miRNA, females n = 5/control and n = 4/$m_{HEa}$-miRNA, original data curated at NCBI/GEO, accession #GSE190017). RNA-seq analysis identified 42 significantly differentially expressed genes (DEGs; FDR-corrected p<0.05), including 15 down-regulated genes (35.71%) and 27 upregulated genes (64.29%) (Fig 4A). To assess for sex-specific response to prenatal $m_{HEa}$miRNA exposure, DEseq2 analysis was conducted separately for male and female placentas, comparing control versus $m_{HEa}$miRNA for each sex. For analysis of males, 12 DEGs were identified (S1 Table), 6 of which were unique to male placentas only, when compared to the grouped analysis (*Tmc5*, *Dock6*, *Frmpd3*, *Sh3bp1*, *Prl7b1*, *Nr5a2*). For analysis of females, 1 DEG was identified (S2 Table), and it was unique to this female only analysis compared to the grouped analysis (*Kif21b*). Pathway analysis was conducted on the 42 DEGs utilizing *ReactomePA* and the KEGG database. Only the Notch signaling pathway (mmu04330) was identified as significantly perturbed (Gene Ratio 3/13, q-value = 0.00521), with transcripts for *Dll4*, *Rfng*, and *Hey1* being significantly upregulated (Fig 4B).

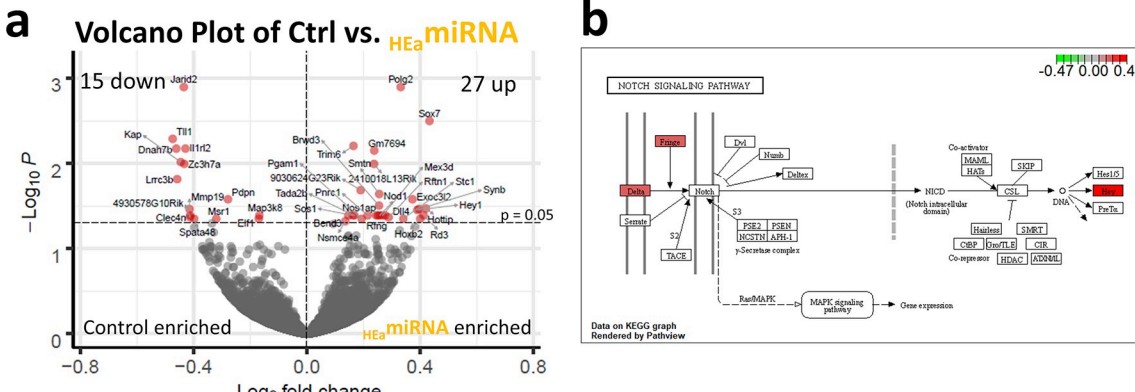

**Fig 4. Prenatal exposure to $m_{HEa}$miRNAs results in differential gene expression and perturbed Notch signaling in the placenta.**
(**a**) Volcano plot of log2 fold change and −log10 p-value of all genes differentially expressed in placentas of $m_{HEa}$miRNA fetuses vs. control. (**b**) KEGG graphic of Notch Signaling Pathway depicting genes dysregulated by prenatal $m_{HEa}$miRNA exposure in pathway. Control male n = 4, Control female n = 5, $m_{HEa}$miRNA male n = 6, $m_{HEa}$miRNA female n = 4.

We also assessed whether other crucial cellular processes such as cell cycle progression and apoptosis were affected in the long term by prenatal $m_{HEa}$miRNA exposure as our previous data did find an impact in an *in vitro* model over a short time course [20]. Gene lists associated with S-phase and G2/M-phase of the cell cycle were extracted from the *Seurat* package in R. A 2-way ANOVA, comparing the averages of z-scores across all relevant genes, showed no significant difference in genes associated with S phase and G2/M (S3A, S3B Fig). In agreement with this, no significant difference was observed in positive and negative regulators of cell cycle progression (S3C–S3H Fig). To determine whether developmental $m_{HEa}$miRNA exposure potentially influenced sensitization for apoptosis, we assessed the expression of gene transcripts associated with the execution phase of apoptosis [46–48], specifically the apoptosis-related caspases, including Caspase 2 (CASP2), CASP3/6/7/8/9/10, and the inflammation-related caspases, CASP1/4/5/11/12 [49]. A 2-way ANOVA, comparing the averages of z-scores across all gene transcripts belonging to these classes, showed that there was only trending significance in the apoptosis-related caspases ($F_{(1,15)}$ = 4.062, p = 0.062) and in the inflammation-related caspases ($F_{(1,15)}$ = 4.046, p = 0.063) due to $m_{HEa}$miRNA exposure. We observed no main effect of sex or interaction effect between treatment and sex for the apoptosis-related caspases and the inflammation-related caspases.

Previous work suggested that $_{HEa}$miRNA exposure has the potential to alter trophoblast differentiation *in vitro* [20]. Therefore, we assessed the expression of genes associated with different cell lineages within the placenta to determine if the influence of $m_{HEa}$miRNAs on the maturation of trophoblasts would have long-term repercussions in our model. We utilized publicly available single cell RNAseq (scRNAseq) data [45] to obtain gene expression patterns that defined cell lineages specific to the non-decidual portion of the placenta (e.g., extravillous trophoblast, fibroblasts, Hofbauer cells, syncytiotrophoblast, and villous cytotrophoblast). However, for all non-decidual cell types assessed, we observed no main effect of treatment, main effect of sex, nor interaction effect between treatment and sex (S3 Table).

### Weighted gene co-expression network analysis reveals a gene network module that is correlated with fetal growth outcomes

Previously, we found that prenatal $m_{HEa}$miRNA exposure resulted in impaired fetal growth and placental size [20]. Specifically, fetal weight, placental weight, crown-rump length (CRL),

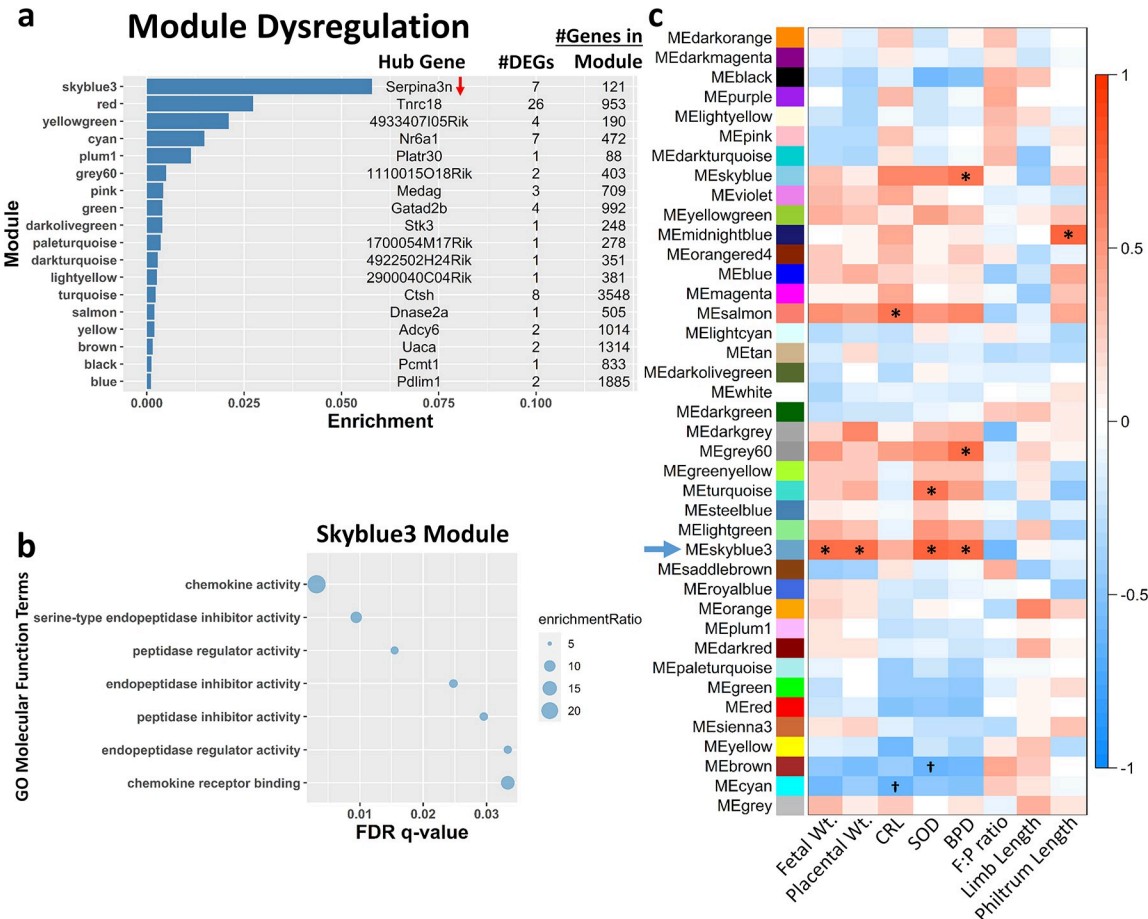

**Fig 5. WGCNA reveals gene network associated with chemokine signaling is related to fetal growth outcomes.** (**a**) Bar chart showing the top $m_{HEa}$miRNA-dysregulated gene modules (y-axis). Hub gene, number of differentially expressed genes (DEGs, adjusted p<0.1), and number of genes constituting each module are shown, with arrows indicating direction of $m_{HEa}$miRNA-induced expression change for each significantly altered hub gene. Ratio of module dysregulated genes relative to total number of genes expressed in the module is shown on x-axis. (**b**) Gene ontology (GO) analysis of skyblue3 module genes focused on molecular function. (**c**) Heatmap of module-trait relationships depicting correlations between module eigengenes (ME) and fetal-placenta growth parameters. Significant and trending correlations (Benjamini-Hochberg adjusted p<0.05 and p<0.1 respectively) are marked. The degree of correlation ($r$) is illustrated with the color legend. BPD = biparietal diameter. CRL = crown-rump length. F:P ratio = placental efficiency. SOD = snout-occipital distance. Control male n = 4, Control female n = 5, $m_{HEa}$miRNA male n = 6, $m_{HEa}$miRNA female n = 4. *p < 0.05, †p < 0.10.

snout-occipital distance (SOD), and biparietal diameter (BPD) were all decreased in fetuses prenatally exposed to $m_{HEa}$miRNAs regardless of sex. Therefore, we sought to identify placental gene networks that might be related to these fetal growth outcomes since the placentas used for RNA-seq analysis came from the same fetuses (fetal growth parameters previously published [20]). Using weighted gene co-expression network analysis (WGCNA), we identified 40 modules of gene networks (S4 Table). Using an adjusted p-value <0.1 from our initial DEseq2 results, 18 of these 40 modules contained at least one DEG. The module with the highest DEG enrichment score was also the only module whose hub gene was also differentially regulated (module skyblue3, hub gene *Serpina3a* was downregulated; Fig 5A). Gene ontology of the skyblue3 module revealed genes within this network are associated with chemokine signaling and cellular response to chemokines (Fig 5B, S4 Fig). Moreover, correlation analysis between module eigengenes (Mes) and fetal and placental growth parameters (Fig 5C, S5 Fig) identified the

skyblue3 module as being significantly related to fetal weight (p = 0.032), placental weight (p = 0.047), SOD (p = 0.010), and BPD (p = 0.025) after FDR correction by Benjamini-Hochberg.

## Weighted gene co-expression network analysis reveals gene network module that is correlated with cardiac output of umbilical artery

Because $m_{HEa}$miRNA exposure resulted in a gradual yet persistent decrease in cardiac output through the umbilical artery we conducted a correlation analysis between WGCNA Mes and fetal blood flow was conducted (Fig 6A, S6 Fig) to identify placental gene modules that were sensitive to changes in umbilical blood flow. We identified statistically significant connections between GD12 umbilical arterial VTI and the black (p = 0.02) and purple (p = 0.01) gene modules. We also observed a significant association between and GD18 umbilical arterial VTI and the yellow module (p = 0.02) after Benjamini-Hochberg FDR correction. The black module contained 1 DEG, and the yellow module contained 2 DEGs, while the purple module

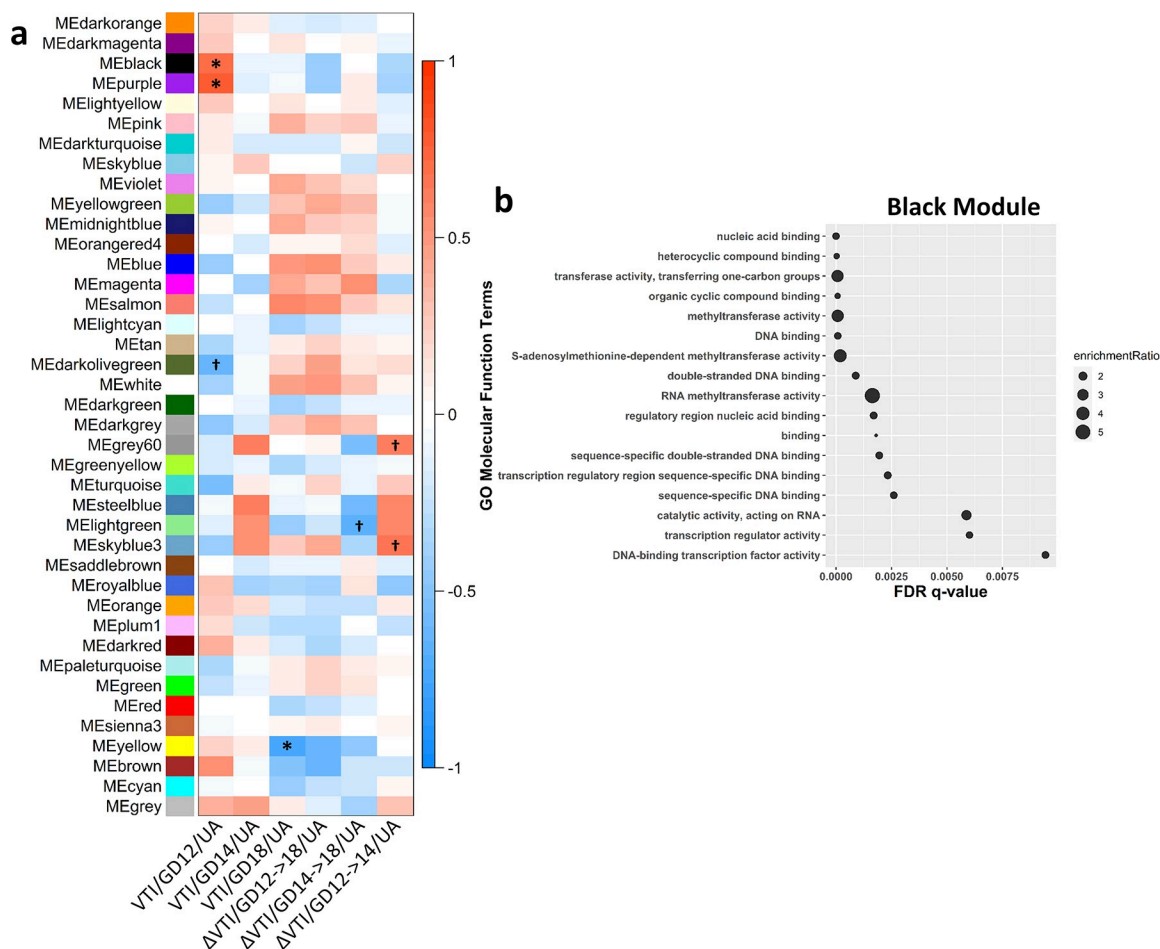

**Fig 6. WGCNA reveals gene network associated with methyltransferase activity and transcription regulation is related to GD12 VTI of the umbilical artery.** (**a**) Heatmap of module-trait relationships depicting correlations between module eigengenes (ME) and umbilical artery VTI over time. Significant and trending correlations (Benjamini-Hochberg adjusted p<0.05 and p<0.1 respectively) are marked. The degree of correlation (*r*) is illustrated with the color legend. (**b**) Gene ontology (GO) analysis of black module genes focused on molecular function. VTI = Velocity Time Integral. UA = umbilical artery. Control male n = 4, Control female n = 5, $m_{HEa}$miRNA male n = 6, $m_{HEa}$miRNA female n = 4. *p < 0.05, †p < 0.10.

contained no DEGs (Fig 5A). Gene ontology of the black module identified methyltransferase activity and transcription regulation (Fig 6B), as cellular processes that were statistically over-represented. For the purple module, DNA-binding transcription factor binding (RNA poly-merase II-specific; S7A Fig), and for the yellow module, small molecule binding (e.g., ion, nucleotide, etc.; S7B Fig) were statistically overrepresented.

Gene ontology analysis was performed on all other modules and results can be found in S5 Table. Additional modules to note as a consequence of these analyses are the orangered4, salmon, and skyblue modules. Genes of the orangered4 module were associated with histone demethylase activity (enrichmentRatio ranging 31.8–163.2). Genes of the salmon module were associated with hormone activity and prolactin receptor binding (enrichmentRatio ranging 4.2–17.9). Genes of the skyblue module were associated with ribosomal RNA (rRNA) binding (enrichmentRatio ranging 25.3–68.1), ubiquitin ligase and transferase inhibitor activity (enrichmentRatio ranging 38.9–45.4), and electron transport and cytochrome-c oxidase activity (enrichmentRatio ranging 9.6–17.9).

## Discussion

This study was based on the premise that a core set of 8 $_{HEa}$miRNAs that are conserved in pla-cental mammals, including humans ($m_{HEa}$miRNAs), when elevated, constitute an environ-mental adaptation, and a presumptive 'danger signature' that predicts adverse birth outcomes over a range of pregnancy complications. Moreover, the effects of these maternal miRNAs were predicted to be directed towards the developing placenta, since the placenta is sensitive to environmental perturbagens such as maternal alcohol exposure [50, 51]. In this study we now report that prenatal exposure to murine $m_{HEa}$miRNAs progressively impairs umbilical cord blood flow over gestation. We showed that this alteration in blood flow is related to growth outcomes and that prenatal $m_{HEa}$miRNA exposure mediated these relationships. Underlying these placental changes were transcriptomic changes in the fetal labyrinthine and junctional zones that identified members of the Notch pathway as increased. Additional WGCNA analy-sis identified correlations between chemokine signaling networks and fetal growth outcomes as well as correlations between placentally directed blood flow and gene networks associated with one-carbon metabolism, DNA-binding transcription factor binding, and small molecule binding. Altogether, these data suggest that by altering placental gene expression, prenatal $m_{HEa}$miRNA exposure alters placental development and subsequently umbilical artery blood flow, resulting in IUGR.

In mice, embryo implantation occurs on GD4.5 with ensuing trophoblast invasion into maternal decidua peaking during GD7.5–9.5 and chorio-allantoic fusion, establishing the early fetal labyrinthine zone on GD9.5–10.5 [52]. Dilation of the spiral arteries occurs between GD10.5–14.5, with diameters increasing by 2.5-fold [53]. Subsequently, extensive expansion of the labyrinthine zone occurs from GD12.5 to term [54], reflecting a greater than 8-fold expan-sion of the fetal compartment capillary volume [55]. If this neovascularization fails to occur normally in the labyrinthine zone, there is the potential for complications such a pre-eclampsia and IUGR [56]. One means by which to measure the impact of impaired placental vasculariza-tion on the fetus is through ultrasound. Umbilical artery ultrasound is used in high-risk preg-nancies to monitor fetal health and is sensitive enough to reduce the risk of perinatal death by 29–38% [57, 58]. We observed that prenatal $m_{HEa}$miRNA exposure attenuated the gestation-related increase in blood flow out of the placenta, culminating in a 14.3% decrease in VTI in the $_{HEa}$miRNA exposed group compared to controls. This is a clinically relevant difference in blood flow, since in studies in human populations, an acute change in VTI from >12% to >15% is indicative of a successful response to treatment in critically ill hypovolemic [59, 60] or

shock patients [61], respectively. In our model, this impaired circulation was chronic and emerged during a critical window of fetal growth and development. It is known that there is a strong association between IUGR and impaired fetal blood flow [62], suggesting the decreased fetal weight and size following prenatal $m_{HEa}$miRNA exposure [20] may be attributable, at least in part, to the impaired umbilical cord blood flow. This hypothesis is supported by partial correlational analyses which showed that prenatal $m_{HEa}$miRNA exposure mediated the relationship between umbilical artery VTI and growth outcomes.

Pathway analysis of genes sensitive to prenatal $m_{HEa}$miRNA exposure identified members of the Notch signaling pathway (*Dll4*, *Rfng*, and *Hey1*) as concomitantly upregulated. Under normal circumstances, hypoxic conditions during the first trimester promote trophoblast migration by inducing Dll4-Notch signaling [63, 64], supporting proper spiral artery development and angiogenesis [65, 66]. As pregnancy progresses and the placenta matures, Notch signaling decreases [67]. However, our data showed increased expression of Notch pathway members in $_{HEa}$miRNA exposed placentas in comparison to controls on GD18, suggesting prolonged activation of the Notch pathway beyond the normal expression window for this pathway. The impaired umbilical artery blood flow that we observed may be a contributing factor, leading to a hypoxic placental environment that promotes Notch signaling. Whether this is a compensatory measure or interferes with placental maturation remains to be determined.

Pathway analysis also revealed chemokine signaling within the placenta as relevant to fetal growth outcomes. It has been shown that prenatal exposures to alcohol and pollution alter maternal serum levels of cytokines and chemokines [68, 69], with repercussions that may last a lifetime. Studies have linked maternal inflammatory status with impaired brain development and neurodevelopmental disorders [68, 70–73]. It is likely that these environmental perturbations also impact placental development and function by altering the finely-tuned communication between fetal-derived trophoblasts and maternally-derived immune cells (e.g. NK cells, monocytic lineage, T cells, as reviewed in Du et al. [74]). Much of this communication is mediated by ligand-receptor signaling and disruption in chemokine ligand levels is associated with preterm labor [75] and pre-eclampsia [76, 77]. In our data, the chemokines CCL2, CCL6, and CCL12 were important members of the '*skyblue3*' module correlating with fetal growth outcomes. All three of these chemokines target CCR2 which is highly expressed in maternal decidual stromal cells and maternal leukocytes, such as NK cells [78, 79]. Stimulation of CCR2 may play a protective role under normal conditions by recruiting and maintaining decidual NK cell populations. Decidual NK cells are unique in that they have the distinct function of promoting angiogenesis and spiral artery remodeling within the decidua, as demonstrated in studies using mice that were NK cell-deficient [80, 81], contributing to pre-eclampsia and negative pregnancy outcomes [82]. Our data identified additional important members of the *skyblue3* module correlating with fetal growth outcomes as CXCL1, CXCL2, CXCL3, and CXCL7, which are considered to be angiogenic and are all known to target CXCR2 [83], which is also associated with angiogenesis [84]. Decreased expression of these ligands in the placenta may result in diminished or disorganized angiogenesis, and as mentioned above, contribute to pre-eclampsia and other negative birth outcomes. Moreover, the placenta attempts to mitigate the effects of these changes on the fetus through self-sacrifice by autophagy [85, 86], reflecting the smaller placentas we observed [20] and the relationship between placental weight and chemokine signaling identified here. But these protective measures may become overwhelmed, resulting in the observed fetal consequences of IUGR and diminished head size relating to chemokine signaling.

This study has some limitations. For instance, it is difficult to ascertain the direct mRNA targets of a group of miRNAs acting together, since there are cell-to-cell variations in mRNA

transcription, in the local context of mRNA target sites, competition from RNA binding proteins and other competing endogenous RNAs that may serve as miRNA 'sponges' [87]. Consequently, the cooperativity between miRNAs remains a poorly addressed area in experimental studies. In the past, we have addressed the direct interactions of up to 3 miRNAs on mRNA targets [88], and a few groups have assessed the interactions between two miRNAs [89]. To our knowledge, we are the first research group to assess the cooperative function of such a large group of miRNAs which are phenotypically relevant to fetal growth restriction and appear to have coordinate effects that are distinct from the effects of each individual miRNA member in this cohort [20]. Future studies are focusing on identifying smaller subsets of these miRNAs which broadly replicate the effects of the larger group. Secondly, we focused on the effects of miRNA delivery to the pregnant dam on fetal blood flow. However, it is possible that decreases in fetal blood flow may be a secondary consequence of changes in maternal blood flow which was not investigated in this study. Since these miRNAs were initially identified as a 2nd trimester response to alcohol exposure in pregnant women, it is quite likely that these miRNAs may also influence maternal circulation, since alcohol exposure itself is known to result in impaired nitric oxide-dependent vasodilation in the maternal uterine artery [90].

## Conclusion

Our data show that a sub-population of eight maternal circulating miRNAs that are evolutionarily conserved across placental mammals, including humans, working collectively have the potential to create long-lasting physiological and transcriptomic changes during development that contribute to negative fetal outcomes. A single exposure on GD10 resulted in subsequent attenuation of umbilical cord blood flow through the remaining gestation and persistent placental gene expression changes on GD18. We found that fetal and placental weight, as well as head size, were related to these alterations in blood flow dynamics and transcription, suggesting an underlying etiology mediated by prenatal $m_{HEa}$miRNA elevation that leads to IUGR. Our work suggests that a better understanding of the role of maternal miRNAs during placental development and angiogenesis will provide additional perspective on gestational pathologies and potentially lead to effective avenues for intervention through the manipulation of circulating miRNAs.

## Supporting information

**S1 Fig. Blood flow acceleration of the umbilical artery in fetal mice.** Results are expressed as the mean ± SEM, Control n = 10, $m_{HEa}$miRNA n = 12.
(TIF)

**S2 Fig. Prenatal $m_{HEa}$miRNA exposure alters relationship between blood flow and fetal growth outcomes.** Heatmap of Pearson correlation matrices for uncorrected analyses (**a**) and analyses correcting for prenatal $m_{HEa}$miRNA exposure (**b**). For all correlation matrices, color intensity indicates a stronger correlation, with red representing a positive relationship and blue representing a negative relationship. BPD = biparietal diameter. CRL = crown-rump length. F:P ratio = placental efficiency. SOD = snout-occipital distance. VTI = Velocity Time Integral. UA = umbilical artery. Control male n = 4, Control female n = 5, $m_{HEa}$miRNA male n = 6, $m_{HEa}$miRNA female n = 4.
(TIF)

**S3 Fig. Analysis of cell cycle-associated and cell cycle regulator genes.** Quantification by average z-score of S-phase-associated genes (**a**) and of G2/M-phase-associated genes (**b**). Individual genes were z-scored across samples, and then, average z-score for each sample was calculated and used for analysis. Quantification by average z-score of G1-S transition-positive

regulators (**c**), G1-S transition-negative regulators (**d**), S-phase-positive regulators (**e**), S-phase-negative regulators (**f**), G2-M transition-positive regulators (**g**), and G2-M transition-negative regulators (**h**). Control male n = 4, Control female n = 5, $m_{HEa}$miRNA male n = 6, $m_{HEa}$miRNA female n = 4.
(TIF)

**S4 Fig. KEGG graphic of cytokine-cytokine receptor interactions depicting chemokines dysregulated by prenatal $m_{HEa}$miRNA exposure.** Control male n = 4, Control female n = 5, $m_{HEa}$miRNA male n = 6, $m_{HEa}$miRNA female n = 4.
(TIF)

**S5 Fig. Heatmap of module-trait relationships depicting correlations between module eigengenes (ME) and fetal-placenta growth parameters.** The degree of correlation (*r*) is illustrated with the color legend and is listed at the top of each square. P-values are Benjamini-Hochberg adjusted and are denoted in parenthesis in each square. Control male n = 4, Control female n = 5, $m_{HEa}$miRNA male n = 6, $m_{HEa}$miRNA female n = 4.
(TIF)

**S6 Fig. Heatmap of module-trait relationships depicting correlations between module eigengenes (ME) and umbilical artery VTI over time.** The degree of correlation (*r*) is illustrated with the color legend and is listed at the top of each square. P-values are Benjamini-Hochberg adjusted and are denoted in parenthesis in each square. Control male n = 4, Control female n = 5, $m_{HEa}$miRNA male n = 6, $m_{HEa}$miRNA female n = 4.
(TIF)

**S7 Fig.** Gene ontology (GO) analysis of purple (a) and yellow (b) module genes focused on molecular function. Control male n = 4, Control female n = 5, mHEamiRNA male n = 6, mHEamiRNA female n = 4.
(TIF)

**S1 Table.**
(CSV)

**S2 Table.**
(CSV)

**S3 Table.**
(XLSX)

**S4 Table.**
(XLSX)

**S5 Table.**
(CSV)

**S1 Data.**
(XLSX)

**S2 Data.**
(XLSX)

## Acknowledgments

Portions of this research were conducted with high-performance research computing resources provided by Texas A&M University (https://hprc.tamu.edu). The authors

acknowledge intellectual and other support from the Collaborative Initiative on Fetal Alcohol Spectrum Disorders (CIFASD) and its participating Phase IV Principal Investigators, Drs. Alison Noble, Annika Montag, Christie Petrenko, Christina Chambers, Christine Austin, Claire Coles, Cristiano Tapparello, Edward Riley, Jeff Wozniak, Joanne Weinberg, Johann Eberhart, Kazue Hashimoto-Torii, Kenneth Lyons Jones, Masaaki Torii, Mike Suttie, Peter Hammond, Sandra Mooney, Sarah Mattson, Scott Parnell, Susan Smith, Tatiana Foroud and Thomas Blanchard. CIFASD is funded by grants from the National Institute on Alcohol Abuse and Alcoholism (NIAAA). Additional information about CIFASD can be found at www.cifasd.org.

## Author Contributions

**Conceptualization:** Marisa R. Pinson, Alexander M. Tseng, Christina D. Chambers, Rajesh C. Miranda.

**Data curation:** Marisa R. Pinson, Rajesh C. Miranda.

**Formal analysis:** Marisa R. Pinson, Kirill V. Larin, Rajesh C. Miranda.

**Funding acquisition:** Marisa R. Pinson, Alexander M. Tseng, Kirill V. Larin, Christina D. Chambers, Rajesh C. Miranda.

**Investigation:** Marisa R. Pinson, Alexander M. Tseng, Tenley E. Lehman, Karen Chung, Jessica Gutierrez, Rajesh C. Miranda.

**Methodology:** Alexander M. Tseng, Kirill V. Larin, Christina D. Chambers.

**Project administration:** Kirill V. Larin, Christina D. Chambers, Rajesh C. Miranda.

**Resources:** Rajesh C. Miranda.

**Supervision:** Rajesh C. Miranda.

**Validation:** Marisa R. Pinson, Rajesh C. Miranda.

**Visualization:** Marisa R. Pinson, Kirill V. Larin.

**Writing – original draft:** Marisa R. Pinson, Tenley E. Lehman, Karen Chung, Jessica Gutierrez, Kirill V. Larin, Christina D. Chambers, Rajesh C. Miranda.

**Writing – review & editing:** Marisa R. Pinson, Alexander M. Tseng, Jessica Gutierrez, Kirill V. Larin, Christina D. Chambers, Rajesh C. Miranda.

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
