## [Decision Letter · Decision Letter 0]

14 Jun 2023

PONE-D-23-14055Maternal Circulating miRNAs Contribute to Negative Pregnancy Outcomes by Altering Placental Transcriptome and Fetal Vascular DynamicsPLOS ONE

Dear Dr. Miranda,

Thank you for submitting your manuscript to PLOS ONE. After careful consideration, we feel that it has merit but does not fully meet PLOS ONE’s publication criteria as it currently stands. Therefore, we invite you to submit a revised version of the manuscript that addresses the points raised during the review process.

We look forward to receiving your revised manuscript.

Kind regards,

Giovanni Tossetta, Ph.D

Academic Editor

PLOS ONE

Journal Requirements:

- https://doi.org/10.1111/acer.14846

In your revision ensure you cite all your sources (including your own works), and quote or rephrase any duplicated text outside the methods section. Further consideration is dependent on these concerns being addressed.

“Portions of this research were conducted with high-performance research computing resources provided by Texas A&M University (https://hprc.tamu.edu). This work was supported by grants from the NIH, R01 AA024659 (RCM), R01 HD086765 (KVL,RCM), U01 AA014835 (CC), F30 AA027698 (MRP), and F31 AA026505 (AMT).**”**

“). This work was supported by grants from the NIH, R01 AA024659 (RCM), R01 HD086765 (KVL,RCM), U01 AA014835 (CC), F30 AA027698 (MRP), and F31 AA026505 (AMT). The funders had no role in study design, data collection and analysis, decision to publish, or preparation of the manuscript.”

Reviewers' comments:

Reviewer's Responses to Questions

**Comments to the Author**

1. Is the manuscript technically sound, and do the data support the conclusions?

Reviewer #1: Partly

Reviewer #2: Yes

2. Has the statistical analysis been performed appropriately and rigorously? 

Reviewer #1: Yes

Reviewer #2: Yes

3. Have the authors made all data underlying the findings in their manuscript fully available?

Reviewer #1: Yes

Reviewer #2: Yes

4. Is the manuscript presented in an intelligible fashion and written in standard English?

Reviewer #1: Yes

Reviewer #2: Yes

5. Review Comments to the Author

Reviewer #1: In this paper, Pinson et al. showed that a administration of pooled eight miRNAs to pregnant mice on gestation day 10 attenuates umbilical cord blood flow. Further RNAseq analysis of placenta from those fetuses demonstrated transcriptomic changes on the Notch pathway. The authors conclude possible interactions between placental gene expression changes and impaired umbilical cord blood flow. These results might explain correlation of miRNAs and Fetal Alcohol Spectrum Disorders. However, the following questions raised:

1) It is unclear what is the target of the miRNAs and why the authors continue to use whole eight miRNAs. It is important to explain attempts to reduce the elements.

2) To understand the experimental systems, it is necessary to show schematic time course diagram of experiments. Such as GD10, injection; GD12 and GD14, Doppler imaging; GD18, imaging, tissue sampling for RNAseq, and so on.

3) Authors found attenuation of umbilical cord blood flow by pooled miRNA injection. However, causes of attenuation is unclear. It is interesting to know whether dam’s blood follow or placenta of fetuses cause abnormal umbilical artery. Thus, it might be important to analyze blood pressure of dam and histology of placenta.

Minor points:

Figure legends are not enough to explain experimental systems. For instance, Figure 1 may include explanation for Doppler imaging.

Number of references should be selected and reduced. For instance, reference 21 to 31 should be representative ones or a review article.

Reviewer #2: In this article the authors evaluated the role of 8 mammalian-conserved MicroRNAs in placental and fetal development using mouse as a model of pregnancy in vivo. These mRNAs appeared dysregulated in pregnancy affected by gestational and fetal diseases such as Preeclampsia or IUGR. Interestingly they demonstrated that prenatal mRNA elevation could alter the expression of placental genes correlated with angiogenesis leading to an impaired umbilical cord blood flow and subsequently, IUGR.

The manuscript is generally well written so it coan be accpetted in the present form.

6. PLOS authors have the option to publish the peer review history of their article (what does this mean?). If published, this will include your full peer review and any attached files.

Reviewer #1: No

Reviewer #2: No

---

## [Author Response · Author response to Decision Letter 0]

10 Aug 2023

Dear Editor, we thank the reviewers for their thoughtful suggestions and comments. Based on these, we have made a number of changes in the manuscript. We also addressed comments and requests made by the editor. We are including reviewer comments (in italics) below followed by our response.

Reviewer #1: 

Reviewer #1: In this paper, Pinson et al. showed that a administration of pooled eight miRNAs to pregnant mice on gestation day 10 attenuates umbilical cord blood flow. Further RNAseq analysis of placenta from those fetuses demonstrated transcriptomic changes on the Notch pathway. The authors conclude possible interactions between placental gene expression changes and impaired umbilical cord blood flow. These results might explain correlation of miRNAs and Fetal Alcohol Spectrum Disorders. However, the following questions raised:

1) It is unclear what is the target of the miRNAs and why the authors continue to use whole eight miRNAs. It is important to explain attempts to reduce the elements.

- The reviewer raises an important issue. In our previous publication (Tseng et al., 2019, DOI: 10.26508/lsa.201800252), we did attempt to assess the function of each individual miRNA and found that they did not explain the activity of the group as a whole. We also conducted initial studies to create functional groupings of miRNAs. We have recently received funding to systematically reduce the number of miRNAs to a core group, and those studies are ongoing. In the meanwhile, to acknowledge the importance of this issue, we have included the following as a study limitation, (line 426), “For instance, it is difficult to ascertain the direct mRNA targets of a group of miRNAs acting together, since there are cell-to-cell variations in mRNA transcription, in the local context of mRNA target sites, competition from RNA binding proteins and other competing endogenous RNAs that may serve as miRNA ‘sponges’ 87. Consequently, the cooperativity between miRNAs remains a poorly addressed area in experimental studies. In the past, we have addressed the direct interactions of up to 3 miRNAs on mRNA targets 88, and a few groups have assessed the interactions between two miRNAs 89. To our knowledge, we are the first research group to assess the cooperative function of such a large group of miRNAs which are phenotypically relevant to fetal growth restriction and appear to have coordinate effects that are distinct from the effects of each individual miRNA member in this cohort 20. Future studies are focusing on identifying smaller subsets of these miRNAs which broadly replicate the effects of the larger group.”

2) To understand the experimental systems, it is necessary to show schematic time course

diagram of experiments. Such as GD10, injection; GD12 and GD14, Doppler imaging; GD18,

imaging, tissue sampling for RNAseq, and so on.

- We included a schematic diagram in Figure 1a

3) Authors found attenuation of umbilical cord blood flow by pooled miRNA injection.

However, causes of attenuation is unclear. It is interesting to know whether dam’s blood

follow or placenta of fetuses cause abnormal umbilical artery. Thus, it might be important to

analyze blood pressure of dam and histology of placenta.

- We agree with the reviewer that this is limitation of our study. We did not measure maternal circulation. Therefore, we included as a limitation, the following statement, (line 437) “ Secondly, we focused on the effects of miRNA delivery to the pregnant dam on fetal blood flow. However, it is possible that decreases in fetal blood flow may be a secondary consequence of changes in maternal blood flow which was not investigated in this study. Since these miRNAs were initially identified as a 2nd trimester response to alcohol exposure in pregnant women, it is quite likely that these miRNAs may also influence maternal circulation, since alcohol exposure itself is known to result in impaired nitric oxide-dependent vasodilation in the maternal uterine artery 90.”

Minor points:

Figure legends are not enough to explain experimental systems. For instance, Figure 1 may

include explanation for Doppler imaging.

- We have increased the explanation for Figure 1

Number of references should be selected and reduced. For instance, reference 21 to 31

should be representative ones or a review article.

- We eliminated the references.

Reviewer #2: In this article the authors evaluated the role of 8 mammalian-conserved MicroRNAs in placental and fetal development using mouse as a model of pregnancy in vivo. These mRNAs appeared dysregulated in pregnancy affected by gestational and fetal diseases such as Preeclampsia or IUGR. Interestingly they demonstrated that prenatal mRNA elevation could alter the expression of placental genes correlated with angiogenesis leading to an impaired umbilical cord blood flow and subsequently, IUGR. The manuscript is generally well written so it can be accepted in the present form.

- We thank the reviewer for their assessment.

Editors Questions:

Style Requirements.

- We have reformatted the manuscript.

Minor occurrence of overlapping text with the following previous publication(s), which needs to be addressed: https://doi.org/10.1111/acer.14846

- We ran a check between the previous paper and this submitted one to compare them, and find only overlap in the methods section. The previous paper dealt with the effects of prenatal alcohol exposure, while this paper deals with maternal miRNA exposure. The workflow of the two papers was nearly identical, but the purpose and outcomes were very different. We have now cited that pervious paper in the current manuscript. On line 96, we say, “General study methodologies and paradigms are as previously published 25.”

Please remove any funding-related text from the manuscript and let us know how you would like to update your Funding Statement.

- We have removed the funding statement from the manuscript. The current funding statement that you have is correct.

We note that you have stated that you will provide repository information for your data at acceptance.

- As we mentioned on the cover page of the manuscript, “RNAseq expression data generated during this study are available in the NCBI/GEO database (accession number, GSE190017). The following secure token has been created to allow review of record GSE190017 while it remains in private status: mrmpcmmkjrklzav.” Following acceptance, we will make GSE190017 public. 

PLOS requires an ORCID iD

- My ORC iD is: orcid.org/0000-0002-8359-892X. Also, to my knowledge, none of the cited paper has been retracted.

---

## [Editor Report · Decision Letter 1]

13 Aug 2023

Maternal Circulating miRNAs Contribute to Negative Pregnancy Outcomes by Altering Placental Transcriptome and Fetal Vascular Dynamics

PONE-D-23-14055R1

Dear Dr. Rajesh C Miranda,

We’re pleased to inform you that your manuscript has been judged scientifically suitable for publication and will be formally accepted for publication once it meets all outstanding technical requirements.

Kind regards,

Giovanni Tossetta, Ph.D

Academic Editor

PLOS ONE

---

## [Editor Report · Acceptance letter]

21 Sep 2023

PONE-D-23-14055R1 

Maternal Circulating miRNAs Contribute to Negative Pregnancy Outcomes by Altering Placental Transcriptome and Fetal Vascular Dynamics 

Dear Dr. Miranda:

I'm pleased to inform you that your manuscript has been deemed suitable for publication in PLOS ONE. Congratulations! Your manuscript is now with our production department. 

Kind regards, 

on behalf of

Dr. Giovanni Tossetta 

Academic Editor

PLOS ONE